# Gender and Renal Insufficiency: Opportunities for Their Therapeutic Management?

**DOI:** 10.3390/cells11233820

**Published:** 2022-11-29

**Authors:** Tiziana Ciarambino, Pietro Crispino, Mauro Giordano

**Affiliations:** 1Internal Medicine Department, Hospital of Marcianise, ASL Caserta, 81031 Caserta, Italy; 2Emergency Department, Hospital of Latina, ASL Latina, 04100 Latina, Italy; 3Department of Advanced Medical and Surgical Science, University of Campania, Luigi Vanvitelli, 80138 Naples, Italy

**Keywords:** renal insufficiency, gender/sex differences, hormones, kidney injury

## Abstract

Acute kidney injury (AKI) is a major clinical problem associated with increased morbidity and mortality. Despite intensive research, the clinical outcome remains poor, and apart from supportive therapy, no other specific therapy exists. Furthermore, acute kidney injury increases the risk of developing chronic kidney disease (CKD) and end-stage renal disease. Acute tubular injury accounts for the most common intrinsic cause of AKI. The main site of injury is the proximal tubule due to its high workload and energy demand. Upon injury, an intratubular subpopulation of proximal epithelial cells proliferates and restores the tubular integrity. Nevertheless, despite its strong regenerative capacity, the kidney does not always achieve its former integrity and function and incomplete recovery leads to persistent and progressive CKD. Clinical and experimental data demonstrate sexual differences in renal anatomy, physiology, and susceptibility to renal diseases including but not limited to ischemia-reperfusion injury. Some data suggest the protective role of female sex hormones, whereas others highlight the detrimental effect of male hormones in renal ischemia-reperfusion injury. Although the important role of sex hormones is evident, the exact underlying mechanisms remain to be elucidated. This review focuses on collecting the current knowledge about sexual dimorphism in renal injury and opportunities for therapeutic manipulation, with a focus on resident renal progenitor stem cells as potential novel therapeutic strategies.

## 1. Background

Acute kidney injury (AKI) is not a self-limiting event but growing evidence in the literature suggests that any injury leads to progression to varying degrees of chronic kidney disease (CKD) [1,2]. The pathophysiological reasons, along with the cellular and molecular mechanisms, underlying the ability of a single, apparently self-limiting, acute event to drive the progression of CKD have yet to be explained. Furthermore, previous studies [3,4,5] have shown that AKI severity, duration, and frequency are associated with an increased risk of progression of CKD. Traditionally, female sex has been included as a risk factor in models developed to predict the risk of AKI developing into CKD associated with cardiovascular surgery, nephrotoxicity from the use of aminoglycoside antibiotics, rhabdomyolysis, and administration of contrast media to perform radiological examinations. A meta-analysis that pooled a set of studies providing gender-stratified incidence of AKI demonstrated that the female sex plays a protective role with respect to the onset of AKI. The results of this meta-analysis have induced the scientific world to reconsider the well-established evidence that female sex is a significant risk factor for AKI. Acute ischemic tubular necrosis is often the most frequent aspect of AKI, and it is in this context that the renoprotection offered by the female sex appears to be more robust [6]. Furthermore, a study based on a population of hospitalized patients with a sample size three times larger than all of the cohorts previously considered by traditional publications examined the need for dialysis in the later stages of the decline of renal function. The study reported that the male sex is associated with an increased incidence of hospital AKI, re-evaluating the protective role of the female gender in AKI and verifying a greater use of dialysis in males compared to their female counterparts [7]. Some data [8,9,10] confirm that the female sex is more protected from kidney damage and suggest the protective role of female sex hormones as an explanation for this phenomenon, while others highlight the harmful effect of male hormones in contributing to kidney damage. Although the role of sex hormones is evident, the underlying mechanisms by which these substances exert their protective role in women and their expository role in men remain to be elucidated. This review focuses on collecting the current knowledge about sexual dimorphism in AKI, with emphasis on the molecular mechanisms and potential therapeutic strategies.

## 2. Methods

Clinical trials were identified in PubMed until 28 September 2022. The search keywords were gender/sex differences, acute kidney injury, and chronic kidney disease. The studies were selected and their references were reviewed for potential inclusion. Studies written in languages other than English were excluded. Two authors (P.C., T.C.) reviewed all study abstracts. All selected studies were qualitatively analyzed.

## 3. Stages in the Evolution of Kidney Damage

The evolution of acute renal damage into chronic renal damage involves various stages in succession up to the repairing processes of the damage, complete or partial depending on the severity of the damage. These processes follow one another with probabilistic and time-dependent mechanisms through different phases involving:The endothelium;The inflammatory response to endothelial damage;Development of fibrosis;Damage repair and attempts at functional recovery.

From a pathophysiological point of view, microvascular integrity, changes in the behavior of leukocytes and pericytes, the ability to survive, and the attempt to restore tubular cell function are all characteristics of both AKI and chronic renal failure. The main pathological mechanisms that help to explain the transition from AKI to CKD include:(i)Endothelial dysfunction, vasoconstriction, and vascular congestion [11];(ii)Interstitial inflammation and associated infiltration of monocytes/macrophages, neutrophils, T and B cells [12,13];(iii)Fibrosis via myofibroblast recruitment and matrix deposition [14,15];(iv)Tubular epithelial damage and dysregulated repair [16].

Thus, after ischemic injury, loss of nephron mass with hyperfiltration of the residual nephron, activation of the renin-angiotensin system (RAS), systemic hypertension, and subsequent glomerulosclerosis, have been described to pave the way from AKI to chronic renal failure [17,18,19,20]. Regardless of the initial insult, evidence of tubular cell loss and scarring, replacement of collagen, and infiltrating macrophages are associated with further renal functional loss and progression to end-stage renal failure. Experimental models have shown that selective epithelial damage could drive capillary rarefaction, interstitial fibrosis, glomerulosclerosis, and progression to chronic renal failure, confirming a direct role for damaged tubular epithelial cells (TEC) [19]. Therefore, tubular epithelial cells have attracted increasing attention. The interaction between endothelial cells, macrophages and other immune cells, pericytes, and fibroblasts often converge in tubular epithelial cells, which play a central role [19]. Recent evidence has reinforced this concept by showing that damaged tubules respond to acute tubular necrosis through two main mechanisms:The polyploidization of tubular cells;The proliferation of a small population of self-renewing renal progenitors [19].

### 3.1. Endothelial Dysfunction and Vascular Congestion

Endothelial dysfunction is characterized by the loss of the mechanisms that regulate vasodilation and vasoconstriction, mediated by endothelium-dependent activity and exerted by nitric oxide (NO), in response to vasomotor stimuli that involve a variation in systemic blood flow. The endothelial-derived NO is physiologically produced by the activity of eNOS (endothelial nitric oxide synthase) [19,20]. The release of NO involves a multi-level response by reducing the tone of vascular smooth muscle by inducing vasodilation, inhibiting platelet aggregation, and promoting leukocyte activation. In the etiopathogenesis of renal damage, capillary rarefaction and therefore the loss of NO production is traditionally considered an important element linked to the potential progression of chronic renal damage. In fact, the reduction of the oxygen supply leads to an inflammatory reaction, ischemia, and necrosis, making the kidney cells vulnerable, which in the grip of hypoxia lose their function mainly due to a block of mitochondrial activity [19,21]. The inflammatory infiltration resulting from this damage subsequently generates the replacement of functionally active cells with fibrotic tissue [22]. The capillary rarefaction, however, in some way induces a reparative response mediated particularly by the vascular endothelial growth factor (VEGF), an endogenous cytokine produced by epithelial cells and directed to endothelial cells. Its production could constitute a compensatory response to microvascular dysfunctions and morphological changes in the nephron, but the mechanisms governing this process have yet to be known [23,24]. An increase in the production and activity of endothelium-specific transforming growth factor β (TGF-β) and from pericytes of growth factor including PDGF (platelet-derived growth factor), angiopoietin, TGF-β, VEGF, and sphingosine-1-phosphate have been observed to contribute to interstitial fibrosis and, therefore, to chronic damage [25]. Pericytes are cells of undifferentiated mesenchymal origin, with contractile function, partially surrounding the endothelial cells (19). As undifferentiated mesenchymal cellular elements, they can, in response to damage, leave their perivascular site differentiating into myofibroblasts and contributing to both vascular rarefaction and an increase in fibrosis [26].

### 3.2. Interstitial Inflammation and the Role of Associated Infiltration of Monocytes/Macrophages, Neutrophils, T and B Cells

After the initial phase of acute kidney injury, early inflammation is followed by pericytes-mediated infiltration of both residents and infiltrating circulating cells, which, as we have said, contribute to disease progression [27,28]. Taking into account their activity in the regulation of inflammation, neutrophils, due to their paracrine effects on tubular epithelial cells, and macrophages can play an important role as determinants of AKI outcomes. In fact, a subset of regulatory T cells (Treg) can act as an inducer of self-tolerance and suppress inflammation by improving immune homeostasis. The CD4 ^+^ and CD8 ^+^ T lymphocytes could also play a protective role and affect the natural history of AKI although this has only been demonstrated in mouse models [29,30]. Conversely, B lymphocytes and macrophages have been described as elements capable of promoting the evolution of renal damage and making it more severe, contributing to the progression from acute to chronic renal damage [31]. Therefore, an increased presence of macrophages is also related to fibrosis and adverse outcomes [32].

### 3.3. Fibrosis

The interaction between activated macrophages, damaged cells, endothelial cells, and the growth factors connected to the damage is responsible for the development of a pro-fibrotic environment. This environment activates the pericytes to proliferate and evolve into myofibroblasts, further inducing the deposition of the extracellular matrix, the deposition of collagen, fibronectins, and other glycoproteins, and therefore extending renal fibrosis and promoting subsequent CKD evolution [33]. Although recognized as the main prognostic factor for CKD, according to current studies fibrosis is considered to be self-sustaining in its process, and a causal relationship between ECM deposition, fibrosis, and damage has not yet been identified in chronic renal damage [34]. According to some studies, the premature presence of fibrosis after initial damage could be a positive element capable of stimulating healing processes by representing a demarcation zone between damaged cellular elements and those spared or not affected by the damage [35]. Therefore, it would appear that fibrosis is not such an important element in the transition from AKI to CKD and that the progression of renal damage requires further phenomena independent of the apposition of fibrotic tissue. On the basis of experimental models, other authors have shown that selective epithelial damage inducing interstitial fibrosis and glomerulosclerosis led inexorably to the progression to chronic renal failure, confirming a direct role of fibrosis in the loss of renal function [19]. Furthermore, it is necessary to specify that even the role of myofibroblasts is somewhat ambiguous in situations of the evolution of renal damage. This is because if they are naturally the main cellular elements that trigger and condition the severity and extent of the fibrosis, contrarily, they can acquire the ability to produce retinoic acid, supporting the integrity and repair of the epithelium and reducing the pro-inflammatory activity of macrophages [36]. Some studies [37,38] suggest that it is essential to better understand the cellular and molecular processes that underlie fibrosis in AKI for the development of therapeutic strategies to arrest the progression of AKI into CKD. However, it would be useful to first establish the exact role of fibrosis and also to identify the elements or mechanisms that underlie the resistance of renal cells to acute damage.

### 3.4. Tubular Damage and Damage Resistance Capacity

All of the molecular and cellular mechanisms described so far lead to the dysfunction of the tubular epithelial cells, which represents the true cause of the progression of renal damage towards CKD [39]. In fact, the failure to recover the integrity of the tubular structure is directly proportional to the risk of the development and progression of CKD, which in turn is closely linked to the severity of the acute episode [39]. However, the severity of the acute episode is not the only determinant of the progression of kidney damage. It is believed that the age of the affected person may also be important, especially in the processes of survival and resistance of kidney cells as well as the resumption of normal activity [40]. The most acute damage in kidney cells occurs in the tubular epithelial cells of the proximal S3 segment of the outer cord stripe. These cells carry out intense activity and require an enormous metabolic and energy potential that comes only in a limited way from the production of ATP, and therefore exploit energy substrates obtained in anaerobiosis or from alternative metabolic pathways [19]. Precisely for this reason, these cells are extremely vulnerable to hypoperfusion, renal hypoxia, and mitochondrial damage [19]. After acute injury, the kidney can overcome the injury and restore its functionality in the right state [41,42,43]. In fact, after a mild injury event, the kidney may return to a structural and functional state that is indistinguishable from normal. Both the cortex and medulla of the kidney tissue can be senescent, including renal tubular epithelial cells (TECs), podocytes, vascular smooth muscle cells, endothelial cells, and mesenchymal cells, among which renal TECs are the most common cells that undergo senescence [41,42]. This is due to the strong compensatory ability of mature kidney cells, which can quickly re-enter the cell cycle from a resting state within 24 h after injury. If the kidney is severely impaired or damaged, it will cause a variety of pathological changes and interstitial fibrosis [41,42]. It has been observed that the most vulnerable cellular elements, once damaged, undergo a process of dedifferentiation. This allows them to return to an undifferentiated state and therefore able to generate new cellular elements that carry out rapid repair in the case of acute injuries, thus preventing chronic damage [44]. It is known that cellular reprogramming is the process of reverting differentiated cells to the pluripotent state, and the pluripotent state can be differentiated into various functional cells [45]. There are different modes of reprogramming. Cells can first return to a pluripotent state and then differentiate into the desired lineages. Alternatively, one cell type can directly convert into another cell lineage by expressing specific factors [45]. Senescence can be improved by initiating reprogramming, which plays a crucial role in blocking the progression of AKI to CKD. Acute senescence causes the cell cycle to be temporarily blocked, which helps the cells to avoid uncontrolled mitosis and provides more time for DNA repair [46]. Other studies have demonstrated that cell senescence in the process of AKI proliferation and repair is not an irreversible process, and its cell phenotype changes dynamically. In the process of tissue damage and repair, cell senescence after an injury is an important factor leading to cell reprogramming [47,48]. Regeneration of tubular epithelial cells is mainly due to a larger renal progenitor population in the S3 segment of the proximal tubule. This explains the high proliferation of tubular epithelial cells observed in this area, but such activity has also been observed in the S2 segment of the proximal tubule and in other uninjured areas of the nephron [49,50,51]. However, it would seem that the entry of these cells into a phase of senescence does not involve the re-reception and therefore the formation of two new differentiated daughter cells. Nevertheless, they undergo an endoreplication program by which the cells replicate their genome without division, becoming polyploid. Polyploidization increases gene replication in response to increased metabolic demands, constantly maintaining differentiated and specialized cell functions. This allows for hypertrophy and functional recovery [49,52].

The authors reported that the targeting TC (tubular cells) polyploidization after the early AKI phase can prevent the AKI-CKD transition without influencing AKI lethality. Senolytic treatment prevents CKD by blocking repeated TC polyploidization cycles [53].

## 4. Gender Differences in Kidney Damage

Sexual dimorphism in renal injury has been acknowledged since the 1940s, and the possible role of sex hormones has been intensively investigated in the past decades [8]. It has been reported that the pathogenesis, clinical features, and prognosis of renal diseases are different between men and women. In particular, men, especially in old age, may be at a higher risk for AKI, but this association is limited to cases of low estimated glomerular filtration rate [54]. The Clinical Practice Guidelines for Improving Overall Kidney Disease Outcomes (KDIGO) for acute kidney injury reported that the female sex is a susceptibility factor that confers a higher risk of AKI, especially when associated with cardiac surgery, rhabdomyolysis, and nephrotoxic substances or drugs [55].

### 4.1. Sex Differences in Renal Anatomy and Physiology

Gender differences in the etiopathogenesis of renal damage and its subsequent repair begin with the different constitutions and anatomical structures of the kidney. In fact, the size of the organ is greater in humans although the ratio between the length of the kidney and body height are homogeneous between the two sexes. There are conflicting data on the number of functional units as for some studies there are no differences in the number of glomeruli, while for others there is a reduction in the percentage of glomeruli in women in a value until to 15% [8]. Hutchens et al. [56] noted that women have a higher glomerular density relative to organ weight than men. Munger et al. [57] reported that women have a higher renovascular resistance, lower absolute GFR, and lower renal plasma flow than men. These data suggest a functional sexual difference in both baseline renal hemodynamics and renal response to vasodilator stimuli. The expression of endothelial NO synthase (eNOS) and superoxide dismutase was found to be higher in female subjects. Although sexual dimorphism in these proteins has not been documented in humans, they may play an important role in the protective effects against renal injury [8]. Depending on the needs of the organism, a large variety of compounds pass through the cell membrane of the nephrons to reach the tubular level to be removed from the tubular fluid and reabsorbed into the blood, or released into the tubular fluid to be secreted with the urine [8]. This activity appears to be influenced not only by the needs of the organism but also by the different activities of the various tubular transporters. It is necessary to underline that in male rodents, the expression of transport protein complexes with secretory function is higher, while the expression of reabsorption transporters is lower. This indicates that in males, a predominantly secretory activity prevails at the tubular level, while in female subjects a resorbing function is prevalent at the tubular level [8].

### 4.2. Influence of Sex Hormones on Renal Activity

Many aspects of renal function depend on the different hormonal structures existing between the two sexes. Potential mechanisms regulated by sex hormones are:Changes in renal hemodynamics both in response to exogenous and endogenous events;Impaired release of vasoactive factors related to the inflammatory response (transcription factors, pro-inflammatory, and pro-fibrotic cytokines) [58];Production of substances with antioxidant properties [59];Production of substances with a protective role against kidney damage and infections.

The hallmark of a healthy endothelium is the appropriate synthesis of nitric oxide (NO) and subsequent relaxation by vascular endothelial cells in response to a vasodilator stimulus. While NO synthesis and NO mediated dilation are the necessary requirements for normal endothelial function, sex differences in vascular reactivity endothelin-I (ET-I) are mediated, at least in part, by differences in the expression and location of ET-I receptor subtypes. In general, women have a higher proportion of endothelial ET~R than ET-1 receptors in vascular smooth muscle, favoring greater dilation to attenuate systemic vasoconstrictor stimuli. Females have a higher expression of cyclooxygenase-2 and produce more prostaglandins and thromboxane in the renal medullary layer [59]. Furthermore, unlike when the production of prostaglandins is sensitive to physiological changes in testosterone in men, in women there is no variation in relation to the blood concentration of estrogens [60]. Additionally, the production of prostanoids is the basis of many mechanisms of vascular tone control that are influenced by sex [61]. Estrogens significantly contribute to the regulation of vasomotor tone in women and promote the release of nitric oxide (NO) through the endothelial activation of NO synthase (eNOS); while on the contrary, androgens are not able to elevate the release of endothelial NO. Generally, exposure to estradiol in women increases vascular relaxation and endothelial vasodilation by increasing blood flow. Estradiol may also increase sensitivity to vasodilatory factors, such as acetylcholine or prostaglandins, thereby reducing the concentrations required to evoke similar vasodilatory responses [62]. However, after menopause, there is a reduction in the beneficial effect of female sex hormones on the adrenergic receptors, which causes an increased risk of hypertension in postmenopausal women [63]. Therefore, the mechanisms contributing to the greater vascular relaxation by estradiol consist of a greater bioavailability of NO associated with greater adrenergic sensitivity [63].

Regarding the inflammatory and pro-fibrotic response and the production of antioxidants, some observations in experimental animals and in humans have shown that the rate of progression of renal disease is influenced by gender. Deterioration of renal function in patients with chronic renal disease is more rapid in men than in women. Potential mechanisms underlying this phenomenon include sex hormone receptor-mediated effects on glomerular hemodynamics, mesangial cell proliferation, and matrix accumulation, as well as effects on the synthesis and release of cytokines, vasoactive agents, and growth factors. Furthermore, estrogens can exert powerful antioxidant actions in the mesangial microenvironment, which can contribute to the protective effect against kidney damage typical only for the female gender [64]. Administration of estradiol also attenuates vascular responses against acetylcholine in women and estrogen supplementation improves endothelium-related dilation through the increasing level of nitrite/nitrate in women. These findings were not observed in males [59].

The administration of estradiol attenuates glomerulosclerosis and tubulointerstitial fibrosis and therefore has a beneficial effect of limiting the progression of chronic kidney damage [65]. Furthermore, estrogen can be protective during acute infectious processes. In contrast, it has been hypothesized that testosterone has pro-fibrotic and pro-apoptotic properties by promoting the increased activity and sensitivity of renal cells to TNF-α and may worsen the symptoms of an acute inflammatory process due to its immunosuppressive effects, as opposed to females who show a more efficient immune response in the case of severe infections [66,67,68]. In a recent study, numbers from the intensive care unit and hospital mortality were comparable between men and women with AKI related to systemic sepsis, evocating the minor role of gender differences in influencing the clinical course of critically ill patients with declines in renal function linked to sepsis [69].

### 4.3. Apoptosis and Gender

Renal tubular cell death is a key process during AKI and correlates with the loss of renal function [70]. As we have already pointed out, the death of tubular cells is associated at the same time with dedifferentiation and self-duplication of surviving tubular cells [70]. Apoptosis has been described as the leading form of ongoing cell death in AKI. However, over the past decade growing evidence has shown that AKI is also based on the mechanism of regulated necrosis [70]. Apoptosis and necrosis are interconnected, and inhibition of one could activate the other [70]. Overall, regulated necrosis contributes to kidney damage and can potentially guide the transition from AKI to CKD, but it is also the first step in renal parenchyma repair along with apoptosis. Moreover, the processes of apoptosis and programmed necrosis can undergo some gender influences, regulated by hormonal mechanisms. Indeed, testosterone has been shown to promote apoptosis in renal tubular cells by triggering caspase activation, Fas upregulation, and Fas ligand expression [71,72]. It has been suggested that the activation of the phosphatidylinositol 3-kinase/Akt pathway protects organs or cells against acute injury and hypoxia through suppression of the cell death machinery and is a critical mediator of cell survival in several molecular pathways [73]. Some authors have observed that basal Akt levels are higher in female subjects and therefore are highly protective during severe kidney damage. Furthermore, in response to acute injury, the prolonged activation of Akt is elicited by estrogens and is directly proportional to their blood concentration, while it is reduced by testosterone [74,75].

### 4.4. Senescence and Gender

In the kidney, cellular senescence is a process described in several states of renal repair and regeneration, including early AKI, the transition from AKI to CKD, and CKD progression. According to some studies, senescent cells accumulate in kidney diseases, and the activation of senescence mechanisms in tubular cells leads to failure of regeneration after AKI or AKI-to-CKD transition [76,77]. For other authors [78], as seen above, senescence is instead a process aimed at reprogramming the tubular cells subjected to damage in order to contribute to cellular repair. Potentially, this arrest of the cell cycle of the tubular cells after acute damage could provide important protection by preventing the division of potentially damaged cells promoting renal parenchyma repair by having resident renal progenitor cells as precursors, stem cells, and senescent cells [70]. We have observed that the dedifferentiation of renal tubular epithelial cells is a prerequisite for regenerative proliferation after acute injury. During the recovery process, the surviving renal cells dedifferentiated to the mesenchymal phenotype and then proliferated and migrated to replace the lost cells, commonly accepted as a mesenchymal–epithelial transition process [49,52]. Some authors have studied in mouse models the variation of the expression of two markers of tissue remodeling or regeneration, vimentin and proliferating cell nuclear antigen (PCNA), observing a maximum peak after two days of acute kidney damage and demonstrating an intense mitogenic activity. It was also observed that the expression of PCNA and vimentin was higher in male rats than in female rats, correlating this result to greater renal damage [79,80]. These results support the hypothesis that the major tissue remodeling predominant in male rats is associated with the severity of the renal injury. Surprisingly, however, after five days from the damage in the female group a decrease in the two markers and consensually an improvement in tubular function was observed, while in males these markers remained elevated in relation to the persistence of the major tubular disorders. This underlines the gender differences in renal tubular changes and tissue remodeling after acute kidney injury [80]. There is growing evidence that the mitochondria play a central role in reparative mechanisms after AKI, whereas in association with vimentin expression, the activity of the mitochondrial enzyme vimentin and translocator protein (TSPO) is also more pronounced in the proximal tubules of the male rats after 24 h of reperfusion [80]. This greater activity present in the male rat would also seem to be attributable to greater severity of renal damage compared to that of females, without the influence of the different hormonal settings existing between the two sexes [80].

More recently it has been observed that in AKI, transforming growth factor-β1 (TGF-β1), after being released, binds to its receptor and activates downstream SMAD2 and SMAD3 to regulate genes associated with renal fibrosis. Administration of estrogen suppresses TGF-β signaling pathways by promoting the degradation of Smad2 and Smad3, improving the response to kidney damage [81]. However, the role of TGF-β1 in renal AKI is still unclear. Some researchers have found that TGF-β1 plays a protective role in acute injury, while others have shown that TGF-β1 accumulation induces AKI [82,83,84]. In our opinion, considering the evidence described so far, the protective role of female sex hormones in limiting acute kidney damage and limiting the inflammatory and fibrotic activity of the damaged parenchyma seems clear. Evidence regarding gender differences regarding reparative processes seems weaker and currently excludes hormonal involvement in these processes.

## 5. Chronic Kidney Disease and Gender

Gender differences referring to CKD are linked to socio-cultural, epidemiological, clinical, and treatment response factors. Women are the main victims of socio-cultural discrimination, as they show little attention to the disease which often results in a delay in diagnosis, a lesser awareness of its severity and its developmental potential, and a greater reluctance to access control compared to men [85,86]. This has a significant impact on the outcome of CKD linked above all to a late start of appropriate treatment and later use of renal replacement therapy among women [87]. Overall, CKD represents conditions that lead to more death in women than men. In fact, if CKDs are among the top ten causes of death in women, as far as males are concerned, these diseases do not fall into this special classification. Prevalence data also indicate that CKD is more common in females than in males [88,89]. Chronic renal failure resulting from autoimmune kidney disease or infectious urinary tract disease is more prevalent in females while hypertension and diabetes prevail in men. Pregnancy, on the other hand, represents a specific risk factor for women since the onset of conditions such as hypertension during this period predisposes them to chronic kidney disease during pregnancy [90]. The decline in renal function appears to be more inexorable and progressive in men. In fact, in men, chronic renal insufficiency also predisposes them to greater fragility linked in part to the decrease in the concentration of testosterone [88]. It has been shown that testosterone can increase the oxidative stress of renal cells, can further activate the renin-angiotensin system, and aggravate renal fibrosis and glomerulosclerosis. On the other hand, estrogens are neutral on renal function and are sometimes even protective in many renal pathological processes [85].

Regarding the gender differences related to the outcomes of the treatment, we have previously mentioned that the majority of patients who start dialysis early are men more than women. This male predominance is explained by the greater progression of renal biological damage attributed to it but also by a greater awareness of the disease from the earliest stages of onset. Women, on the other hand, especially those in their senior years, preferentially choose to practice conservative treatments, trying to procrastinate the most invasive treatments as much as possible [91,92,93]. This attitude is largely explained by the fact that more and more women live alone, live longer than men, and more easily refuse the help of a caregiver. From a therapeutic point of view, women on renal replacement therapy tend to have more problems than their male counterparts. In fact, women have more difficulty in placing arteriovenous fistulas due to the reduced number of suitable-caliber vessels commonly used for this practice [94]. Therefore, they are usually undergoing dialysis using a venous catheter [95,96]. For this reason, women tend to have many more procedural complications than men, and therefore higher hospitalization rates, worse quality of life, and more serious comorbidities than men [97]. In women, the risk of overestimating the duration and intensity of filtration and of having an excessive overdose of erythropoietin is much more frequent due to the fact that the treatment plans are adapted to the dosages of men [98,99]. The onset and course of chronic renal failure can also be improved by measures that can be directly applied to the daily lifestyle, such as greater control of blood pressure values, serum lipid levels, and the achievement of optimal blood glucose control in the case of the diabetic patient. In patients with CKD, a low-protein dietary regimen can delay the development of kidney damage and postpone the time needed to resort to dialysis therapy [99]. The low protein diet helps to slow down the progression of CKD, especially in women who show greater therapeutic results in the control of proteinuria and blood lipid concentrations [100]. On the contrary, especially in the elderly with type 2 diabetes, the long-term effects of a regimen even with an average protein intake are associated with a decline in renal function, an increase in proteinuria, low-grade inflammation, and greater oxidative stress without an improvement in lean mass values [101]. A meta-analysis provides the same result, observing a modest efficacy of a low-protein diet on the major outcomes of CKD prognosis for patients with diabetic nephropathy [102]. Considering that CKD can last for many years in the diabetic patient, it has been observed that although the decline in renal function is progressive, symptoms related to mood deflection are reduced resulting in a better quality of life [103]. Better efficacy of various low-protein diets could be achieved with sustainable intervention and better compliance of CKD patients. The data concerning the function of the low-protein diet are still uncertain and with weak evidence, especially because this topic has not been addressed in light of any differences in gender. Studies focusing on gender differences in verifying the outcome of low- protein diets and further meta-analysis focusing on the role of gender on the efficacy and safety of these diets are needed, especially in diabetic nephropathy.

## 6. Conclusions

A significant body of evidence supports that the evolution of an acute kidney injury into a chronic kidney injury is characterized by gender differences that originate primarily from different anatomy and physiology in the renal tubule’s absorptive and secretory function between the two sexes. Beyond this difference, it would seem that in particular female subjects are less susceptible to acute kidney damage and are able to restore residual renal function more quickly due to the protective activity of estrogens. In contrast, the harmful effect of male hormones has been demonstrated, although other studies are needed to precisely characterize the molecular mechanisms involved. Less evident is the evidence showing gender differences in the repair and replication processes of damaged renal cells, mostly related to the severity of damage in males. On this last aspect, it will be necessary to concentrate on future studies.

## Data Availability

Not applicable.

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
