# Peer review of "Gender and Renal Insufficiency: Opportunities for Their Therapeutic Management?"

_cells, 2022, doi:10.3390/cells11233820_

Round 1
Reviewer 1 Report
1) On page 8 where you discuss references 89 and 91, the sentences are non-sensical. I suspect that the meaning of the sentence was lost in the editing process.
Author Response
17 November 2022
Dear Prof. Doctor
Please, find enclosed the revised version of the manuscript entitled: “Gender and renal insufficiency: opportunities for their therapeutic management?” We thank the Editor and the Reviewers for their comments and we hope that the following changes will now make the manuscript suitable for publication on the CELLS. Please see the following list of the changes made in manuscript.
Reviewer 1
1) In according to Reviewer’ comments we now have correct on page 8 references 89 and 91.
Thank you
Best regards,
Tiziana Ciarambino
MD, PhD
Reviewer 2 Report
In their paper, Ciarambino T. and collegues aimed to dissert the sexual dimorphism of renal injury and the opportunities for their therapeutic manipulation. The subject is interesting and it falls within the scope of this luornal. While the theme was largely dissected, the paper has a good flow and can add new insights into the field of gender and renal injury. Nevertheless, before recommending this paper for publication, some minor points need to be addressed as follow:
-Authors can could add an interesting recently published article on polyploidy (doi: 10.1038/s41467-022-33110-5, PMID: 36195583).
-there is a problem with the references: I found 2 time the same reference (49 and 77). Moreover different style are mixed.
Author Response
17 November 2022
Dear Prof. Doctor
Please, find enclosed the revised version of the manuscript entitled: “Gender and renal insufficiency: opportunities for their therapeutic management?” We thank the Editor and the Reviewers for their comments and we hope that the following changes will now make the manuscript suitable for publication on the CELLS. Please see the following list of the changes made in manuscript.
Reviewer 2
1) In according to Reviewer’ comments we now add the following sentence “The authors reported that the targeting TC (tubular cells) polyploidization after the early AKI phase can prevent AKI-CKD transition without influencing AKI lethality. Senolytic treatment prevents CKD by blocking repeated TC polyploidization cycles (PMID: 36195583)” in the Tubular damage and damage resistance capacity section.
2) In according to Reviewer’ comments we now correct the reference and delete ref 77 (49 and 77).
3) In according to Reviewer’ comments we now correct different style.
Best regards
Tiziana Ciarambino
MD, PhD